# Impact of SARS-CoV-2 infection and mitigation strategy during pregnancy on prenatal outcome, growth and development in early childhood in India: a UKRI GCRF Action Against Stunting Hub protocol paper

Modou Lamin Jobarteh ,[1] Deepak B Saxena,[2] Bharati Kulkarni,[3] Komal Shah,[2] Santosh Kumar Banjara ,[4] Priyanka Akshay Shah ,[2] Farjana Memon ,[2] Monica Chilumula,[5] Dharani Pratyusha Palepu ,[4] Kiruthika Selvaraj,[3] Teena Dasi,[3] Radhika Madhari,[3] Beatriz Calvo-Urbano,[6] Julie Dockrell,[7] Catherine Antalek,[8] Hilary Davies-Kershaw ,[9] Elaine Ferguson,[1] Claire Heffernan[1,10]

For numbered affiliations see end of article.

**Correspondence to**
Dr Modou Lamin Jobarteh; modou.jobarteh@lshtm.ac.uk

## ABSTRACT

**Introduction** The COVID-19 pandemic has offset some of the gains achieved in global health, particularly in relation to maternal, child health and nutrition. As pregnancy is a period of plasticity where insults acting on maternal environment have far-reaching consequences, the pandemic has had a significant impact on prenatal outcomes, intrauterine and postnatal development of infants. This research will investigate both the direct and indirect impacts of the COVID-19 pandemic during pregnancy on prenatal outcomes, growth and development in early childhood.

**Methods and analysis** Community and hospital data in Hyderabad and Gujarat, India will be used to recruit women who were pregnant during the COVID-19 pandemic and contracted SARS-CoV-2 infection. In comparison with women who were pregnant around the same time and did not contract the virus, the study will investigate the impact of the pandemic on access to healthcare, diet, nutrition, mental health and prenatal outcomes in 712 women (356 per study arm). Children born to the women will be followed prospectively for an 18-month period to investigate the impact of the pandemic on nutrition, health, growth and neurocognition in early childhood.

**Ethics and dissemination** Ethics approval was granted from the institutional ethics committees of the Indian Institute of Public Health Gandhinagar (SHSRC/2021/2185), Indian Council of Medical Research-National Institute of Nutrition (EC/NEW/INST/2021/1206), and London School of Hygiene and Tropical Medicine (72848). The findings of the study will be disseminated to policy and research communities through engagements, scientific conferences, seminars, and open-access, peer-reviewed publication.

## INTRODUCTION

India has an enduring commitment to improve the nutrition and health of its population.[1] Globally, India has one of the highest

**WHAT IS ALREADY KNOWN ON THIS TOPIC**
⇒ The COVID-19 pandemic has caused unprecedented disruptions to health, food systems, economies and the livelihoods of millions of people.
⇒ The pandemic has also contributed to the morbidity and mortality of millions of people globally, and caused unusually higher adverse maternal and birth outcomes, including maternal death, stillbirth and depression.

**WHAT THIS STUDY ADDS**
⇒ This study will improve understanding of the long-term impact of SARS-CoV-2 infection and public lockdown during pregnancy on child growth and development.
⇒ It will investigate the impact of the pandemic on access to healthcare, food and nutrition during pregnancy and maternal mental health.

**HOW THIS STUDY MIGHT AFFECT RESEARCH, PRACTICE OR POLICY**
⇒ This study will support post-pandemic recovery and resilience efforts and help guide the development of evidence-based approaches and strategies to protect women and children.

burdens of undernutrition, with 34.7% of children under-5 years of age having stunted growth and 51.4% of women of reproductive age (ie, 15–49 years) being affected by anaemia,[2] which are directly attributable to factors such as inadequate dietary intake of nutritious foods, infectious diseases, poor hand hygiene and sanitation.[3] These figures

and other indicators of maternal and child undernutrition and mortality are likely to have increased with the emergence of the coronavirus (COVID-19) pandemic.[4] The exacerbation is likely due to disruptions to food systems, earnings and nutrition programmes affecting access, affordability of foods and dietary intake.[5–7]

India has been severely impacted by the pandemic with more than half a million (>530 000 as of January 2023) recorded deaths due to COVID-19.[8] The second wave of the pandemic in India, which started around March 2021, has been the deadliest so far, affecting many sectors of the country including health systems, the economy and food systems.[9] In India and other affected countries, mitigation strategies such as mandatory face covering, public lockdown and curfews were implemented to curb the spread of the virus, and minimise severe forms of the disease, hospital admission and death.[10] While these mitigation strategies are necessary to reduce high infection rates, its introduction can have unintended consequences on high-risk groups such as pregnant women and offspring.

Research has revealed that pregnancy outcomes have worsened worldwide during the pandemic, with an increase in maternal deaths, stillbirths and maternal depression,[11 12] and with adverse effect on early childhood neurocognitive development.[13] During the second wave of the pandemic, hospitals across India reported an unusually high spike in maternal mortality and stillbirths.[14] This was attributed to disruptions in Maternal and Child Health services, in part due to pregnant women not attending routine antenatal clinics for fear of contracting the virus and hospitals struggled to support pregnant women who contracted the virus.[14]

This study will investigate the impact of SARS-CoV-2 infection and public lockdown during pregnancy on growth and development in early childhood in Gujarat and Hyderabad, India. Currently, the data on the impact of SARS-CoV-2 infection or public lockdown during pregnancy on childhood growth and development are very limited. As a result, there is high public health interest to establish research cohorts that can provide an understanding of the long-term impacts of the COVID-19 pandemic during pregnancy.

The evidence generated from this study will support post-pandemic recovery and resilience efforts and help support the development of evidence-based approaches and strategies to protect women and children.

## METHODS AND ANALYSIS
### Study design
This cohort study is designed to longitudinally assess the impact of the COVID-19 pandemic on growth and development in early childhood. The study will comprise of two groups of dyads (mother–baby pairs) through recruiting: (a) women who were pregnant during the COVID-19 pandemic in India and had SARS-CoV-2 infection during the pregnancy, and (b) women who were pregnant around the same time and had no known SARS-CoV-2 infection (i.e., were tested negative) during the pregnancy. Mother–baby pairs in the two groups will be recruited postnatally, between 0 and 24 months after birth, and prospectively followed for an 18-month period. Hospital and community records will be used to match the groups (cases and controls) for age, sex (of the child) and location (community, village, town or city) to facilitate unbiased comparison. Study-specific assessments will be conducted at enrolment into the study (ie, within 1 week of informed consent) and 6, 12 and 18 months later to investigate the impact of SARS-CoV-2 infection and public lockdown during pregnancy on maternal, birth and the development trajectory of children in the two groups. The conceptual framework of the study is elaborated in figure 1.

### Outcomes
The primary outcome of this study is growth and development in early childhood, which includes anthropometry (i.e., length, weight, mid-upper arm circumference (MUAC), head circumference, skin fold thickness and associated Z-scores, including length of age Z-score (LAZ) and weight of length Z-score (WLZ)) and neurodevelopment (ie, cognition, motor and social development) during the 18-month follow-up period. Other outcomes of interest are availability, access and affordability of foods, healthcare and maternal mental health during pregnancy, birth/pregnancy outcome, child morbidity, nutritional status, biochemical markers of inflammation, stress, growth hormone, plasma ACE2 and transmembrane protease, serine 2 (TMPRSS2) (table 1).

### Recruitment
The study will be conducted in Ahmedabad and Sabarkantha regions of Gujarat, and Hyderabad, Telangana, India. In Gujarat, hospital and community records of women who had COVID-19 infection (confirmed through PCR test) during pregnancy will be used to generate the contacts of women to approach for recruitment. In Hyderabad, women recruited in an existing observational study and tested for COVID-19 infection during pregnancy will be identified for inclusion. Women identified for the study will be approached by study staff. At the first point of contact, study staff will use a brief eligibility questionnaire to confirm the identity and eligibility of the woman. The criteria for eligibility are as follows:

► Women aged 18–45 years.
► Pregnant during the coronavirus pandemic in India.
► Had a coronavirus test during pregnancy.
► Singleton pregnancy.
► Baby aged between 0 and 24 months.
► Currently living in and agrees to continue living in areas of Gujarat and Hyderabad during the study period.

Once eligibility is confirmed, staff will use the participant information sheet to explain the details of the study procedures. Women will be given up to 48 hours

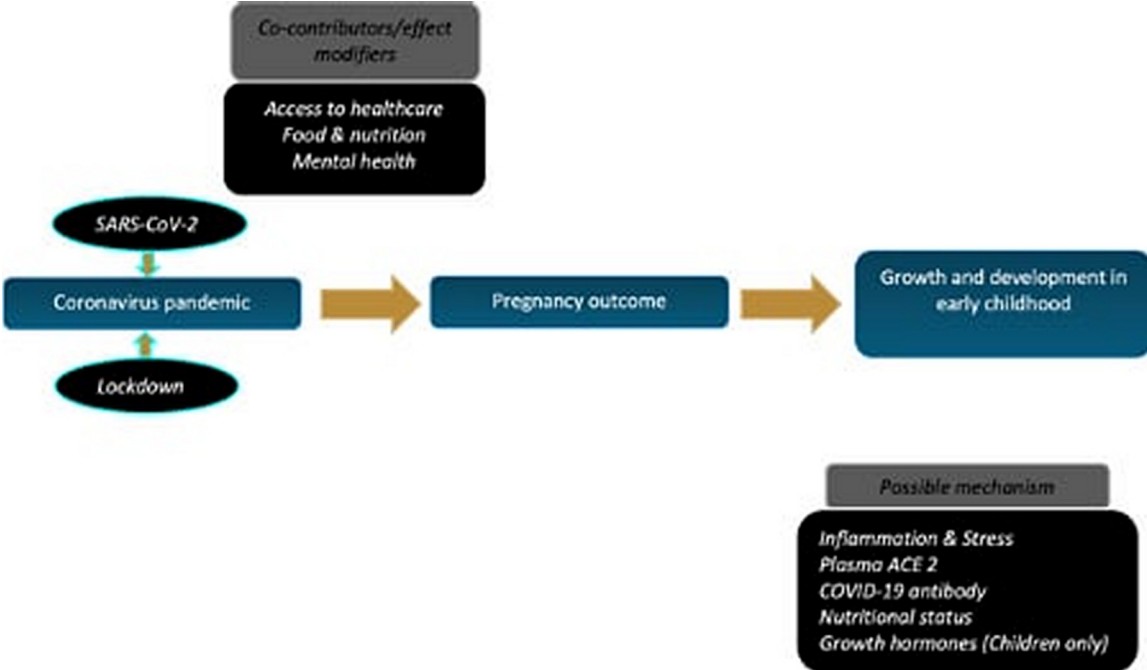

**Figure 1** A conceptual framework of the study. The study will investigate the impact of the pandemic (including independently SARS-CoV-2 infection and public lockdown) on pregnancy outcomes and growth and development in early childhood. In addition, the impact of the pandemic on access to healthcare, food and nutrition and mental health in the women and their role, as co-contributor/effect modifiers, in pregnancy and childhood outcomes will be investigated. Furthermore, the study will evaluate possible explanations of the underlying mechanisms through investigation of biomarkers involved in inflammation, stress, plasma ACE2, nutritional status, etc, in mothers and children.

to decide on participation. Informed consent will be obtained through fingerprint or signature, and women who cannot read or write (illiterate) will require the presence of an impartial witness during the consenting process, who must countersign the consent form. In addition, a control group comprising of women who tested negative for SARS-CoV-2 during pregnancy will be recruited alongside the cases. Case and controls will be matched for age and sex of the child, and address (location). Dyads (mother–baby) recruited into the study will be assigned a unique study identification number which will be used in all future analyses.

### Data and sample collection
#### Maternal questionnaire
At enrolment, a structured questionnaire will be used to investigate the impact of COVID-19 pandemic on access to food, nutrition, healthcare, morbidity and mental health of women during pregnancy (see online supplemental file). It will include questions on the participant's socioeconomic and health status, including marital status, ethnicity, caste, household size, level of education, employment, sources of income, household water sources, energy type, material possessions, and whether they currently have or had been diagnosed with diseases such as diabetes, hypertension, stroke or cancer.

A non-quantitative Food Frequency Questionnaire will be used to explore dietary habits of the women during the pandemic. Participants' usual dietary intake of foods (including staple foods) and sources of these foods will be investigated. We will determine whether the availability, accessibility, affordability and consumption of the foods have changed during the pandemic. Where changes in dietary habits due to the pandemic are reported, prompts will be asked to establish the extent of the changes. For example, if the affordability of basic foods increases, we will ask how much the increment was, and if changes in food consumption are reported, we will find out whether the changes only relate to the quantity of foods/meals consumed or whether the usual foods/diets were substituted for other foods to facilitate satiety and nutrition.

Participants will be asked whether they visited antenatal clinics during pregnancy, and whether access to the clinics and/or services provided by the clinics was disrupted by the pandemic. Questions will be asked about reasons for non-attendance of antenatal clinics. Where antenatal services were provided during the pandemic, we will explore the level of care received and whether it matches the standard antenatal care in India. Furthermore, participants will be asked whether they have received counselling on COVID-19 during pregnancy, and whether the counselling has helped them understand the disease, its impact on pregnancy, vaccination and misinformation.

The impact of the pandemic on maternal mental health will also be investigated. A qualitative questionnaire will be developed and validated in the population (non-study participants) prior to its use. The questionnaire will ask mothers about the impact of the pandemic

**Table 1** Assessments to be conducted in mothers and children at enrolment and follow-up visits

| Timeline | Mother | Child |
|---|---|---|
| Enrolment | *Questionnaire*: household conditions | *Questionnaire*: breastfeeding practice |
| | *Questionnaire*: health and nutrition during pregnancy* | *Anthropometry*: weight, length, MUAC, head circumference, etc |
| | *Anthropometry*: weight, height, MUAC and BMI | *Questionnaires*: cognition (HOME, CREDI, OX-NDA) |
| | *Health records*: SARS-CoV-2 test, date, variant, vaccination status* | *Blood*: Fe, Zn, Ca, Vit A/D, IGF-1, IGF-2, IGFBP3, CRP, AGP, IL-1, IL-6, IL-10, TNF-alpha, ACE2, TMPRSS2, serotonin, tryptophan, SARS-CoV-2 antibody |
| | *Blood*: Fe, Zn, Ca, Vit A/D, IGF-1, IGF-2, IGFBP3, CRP, AGP, IL-1, IL-6, IL-10, TNF-alpha, ACE2, TMPRSS2, serotonin, tryptophan, SARS-CoV-2 antibody | *Diet*: usual dietary intake using 2 non-consecutive 24HDR |
| | *Diet:* usual dietary intake using 2 non-consecutive 24HDR | *Questionnaire*: health and morbidity |
| | | *Health records*: birth outcomes (preterm, SGA, LBW, etc)* |
| 6/12/18-month visits | *Questionnaire*: health and morbidity | *Questionnaire*: breastfeeding practice |
| | *Diet:* usual dietary intake using 2 non-consecutive 24HDR | *Anthropometry*: weight, length, MUAC, head circumference, etc |
| | | *Questionnaires*: cognition (HOME, CREDI, OX-NDA) |
| | | *Blood*: Fe, Zn, Ca, Vit A/D, IGF-1, IGF-2, IGFBP3, CRP, AGP, IL-1, IL-6, IL-10, TNF-alpha, ACE2, TMPRSS2, serotonin, tryptophan, SARS-CoV-2 antibody |
| | | *Diet*: usual dietary intake using 2 non-consecutive 24HDR |

Blood samples will be collected only at enrolment and 6-month visits.
*These are mostly retrospective data. The data will be obtained through antenatal cards, hospital records and verbal interviews with the mothers.
AGP, alpha-1-acid glycoprotein; BMI, body mass index; Ca, calcium; CREDI, Caregiver Reported Early Development Instrument; CRP, C reactive protein; Fe, iron; 24HDR, 24-hour dietary recall; HOME, Home Observation for Measurement of the Environment; IGF, insulin-like growth factor; IGFBP3, insulin-like growth factor binding proteins-3; IL, interleukin; LBW, low birth weight; MUAC, mid-upper arm circumference; OX-NDA, Oxford Neurodevelopment Assessment; SGA, small-for-gestational age; TMPRSS2, transmembrane protease, serine 2; TNF, tumour necrosis factor; Zn, zinc.

during pregnancy on components of their mental health including stress, anxiety, fatigue, depression and insomnia. We will further investigate the risk factors to the mental health of the women during the pandemic, whether it was related to bereavement, isolation, lockdown, worry/fear of getting infected with COVID-19, fear of losing their unborn child, fear of dying, fear of losing a family member, fear of not having enough food, loss of income and inability to attend antenatal clinics. Women can give as many self-reported underlying risk factors as applicable, but we will seek to establish their top three risk factors.

### Pregnancy/birth outcome

Antenatal cards, hospital records and interviews will be used to collect data on pregnancy outcomes such as pre-eclampsia, gestational diabetes, gestational age, preterm birth, congenital deformities and delivery type (vaginal–natural or artificially induced labour, and caesarean section). In addition, health records of neonatal outcomes such as birth weight, placenta weight, infant length, head circumference and MUAC will be collected.

### Breastfeeding practice

At enrolment, a questionnaire will be used to explore whether breastfeeding practices during the pandemic might have differ from standard practice. The women will be asked whether early initiation of breast feeding (usually within 1 hour of birth of the baby to benefit from colostrum breastmilk) was conducted soon after the birth of the baby. Mothers will be asked whether the baby is currently breast feeding, and if not, when was the baby weaned and age when complementary feeds including drinks and/or water were introduced. This questionnaire will provide understanding of the behaviours around adherence to common breastfeeding practices such as exclusive breast feeding under 6 months, continued breast feeding at 1 year, and the stage of introduction of solid, semisolid, and soft food.

## Biochemical analyses

Biomarkers of inflammation (C reactive protein (CRP), alpha-1-acid glycoprotein (AGP), interleukin (IL)-1, IL-6, IL-10 and tumour necrosis factor (TNF)-alpha), growth (insulin-like growth factor (IGF)-1, IGF-2 and insulin-like growth factor binding proteins-3 (IGFBP3)), nutrition (ferritin, calcium, vitamin A, vitamin D and zinc), SARS-CoV-2 receptor proteins (ACE2, TMPRSS2) and COVID-19 antibody will be investigated in mothers and children. These biomarkers will facilitate understanding of the long-term/residual effect of the pandemic on critical physiology, susceptibility to the virus and inter/counteraction between the physiological markers (including between nutrients, vitamin D, zinc, iron and markers of SARS-CoV-2 entry into host cell/severity such as ACE2 and TMPRSS2).

A nurse trained in phlebotomy will collect 5 mL of venous blood samples from the children at enrolment and at 6-month follow-up visit, and 10 mL of blood sample from the mothers at enrolment only. The samples will be used to estimate nutritional status (iron, zinc, calcium, vitamins A and D), growth factors (IGF-1, IGF-2, IGFBP-3), inflammation markers (CRP, AGP, cytokines (IL-1, IL-6, IL-10, TNF-alpha)) and markers related to COVID-19 susceptibility (ACE2, TMPRSS2, tryptophan, serotonin) and COVID-19 antibody.

## Anthropometry

Growth of children will be assessed through anthropometric measurements of weight, length, knee-to-heel length, head circumference, MUAC and skin fold thickness (biceps, triceps and subscapular) at enrolment and follow-up visits. All measurements will be conducted using equipment that is regularly calibrated. Measurements will be conducted in triplicate. The measurements will be used to determine Z-scores using the WHO Growth Standards for LAZ, WLZ, triceps-for-age and subscapular-for-age, and to classify the children as stunted or not stunted (LAZ <−2 SD) and wasted or not wasted (WLZ <−2 SD). The linear growth rate of children will also be determined

In addition, the height, weight and MUAC of mothers will be measured only at enrolment and BMI (body mass index) calculated. Where possible, secondary data on weights of the women during pregnancy will be recorded.

## Neurocognitive development

Age-appropriate measurements will be used to investigate the impact of the pandemic on cognitive, motor, language, emotional and social development of children. Three methods will be used to measure both the environment in which childhood development is taking place and the development itself (see online supplemental file).

Home Observation for Measurement of the Environment (HOME)[15] will be used to assess the developmental environment at both visits. HOME assesses the emotional support and cognitive stimulation children receive through their home environment, planned events and family surroundings. HOME contains both observational (by an assessor) and parent/caregiver-reported questionnaire items. The observational items of the assessment cover domains such as the quality of parent–child interactions, cleanliness and order of the home, and distinct features of the dwelling. Self-report items for parents include questions about their child's activities over the past days or weeks, discipline and parent–child interactions.

Caregiver Reported Early Development Instrument (CREDI)[16] and Oxford Neurodevelopment Assessment (OX-NDA)[17] for infants will be used to assess early childhood neurocognitive development. CREDI measures development in five inter-related domains (motor, language, cognition, socioemotional and mental health). CREDI will be administered by assessors to parents/caregivers to report milestones and behaviours that are easy for caregivers to understand, observe and describe. OX-NDA measures several areas of development including cognition, fine and gross motor skills, expressive and receptive language, behaviour (positive, negative and global), executive function, empathy, problem-solving, attention and socioemotional reactivity. OX-NDA assessment consists mainly of observation items which will be completed primarily by an assessor with some questions completed by the parent/caregiver.

## Dietary assessment

Multipass, quantitative 24-hour dietary recall method will be used to estimate the food and nutrient intake of mothers and children at enrolment and follow-up visits. Staff trained in the multipass method will visit the homes of participants to complete the dietary questionnaire. The dietary intake of mothers and children will be assessed separately. A dietary application will be used to enter recalls of all the foods and drinks mothers and children consumed in the preceding 24 hours. Food portion sizes will be estimated using real foods/food models whenever feasible, photo album, cost equivalent or number (discrete commercial food such as biscuits) or size (small, medium, large). Food intake data will be translated into intake of energy and nutrients using the Indian food composition table.[18]

## Health and morbidity questionnaire

A health and morbidity assessment questionnaire will be used to investigate the burden of disease in mothers and children. At enrolment and follow-up visits, mothers will be asked whether she or her child has had any illness over the last preceding months, and if any medication was administered due to the illness. In case of a hospital admission, information on diagnosis and treatments received while at hospital will be collected. In addition, records of childhood vaccination the children have received will be collected.

## Analysis plan

Sample size calculations were based on LAZ at 18 months of childhood and conducted using STATA with

the command 'power two means −1.5 −2.0, SD1(1.5) SD2(1.5) power (0.8) alpha (0.05)'. The power of the study and the significance level were set at 80% and 5%, respectively. Mean LAZ in children born to non-infected women was assumed to be −1.5 and we were interested in assessing a minimum difference of 0.5 in LAZ between children in the infected and uninfected groups. The SD of LAZ in both groups was assumed to be 1.5. After considering an attrition rate of 20%, the final sample size was calculated at 712 (ie, 356 participants per group).

Characteristics of pregnant women at the time of recruitment will be summarised and presented by cohort. The summaries will include medians and IQRs in the case of continuous variables, and the number of individuals per category and their percentages in the case of categorical variables. The WHO R package (anthro)[19] will be used to calculate LAZ and WLZ at enrolment and at 6-month follow-up. The analyses will be carried out in R using a Bayesian framework by means of JAGS[20] or Stan.[21] Predictor variables will be standardised to facilitate model convergence and the selection of prior distributions. The parameters' posterior distributions, mean and 95% credible intervals will be computed and compared. Models will be compared with tools such as the Deviance Information Criterion and their Widely Applicable Information Criterion.

Generalised linear mixed models (GLMMs) will be used to evaluate the impact of COVID-19 on children's quantitative outcome variables. GLMMs can account for repeated measures per child over time, by modelling children as random effects. COVID-19 infection status during pregnancy will be modelled as an index variable. Adjusted and unadjusted models will be derived. The latter will include sex of child and study site. Furthermore, the effect of other covariates on children's outcome variables will be investigated by means of univariate and multivariate regression modelling. These covariates include household socioeconomic status, maternal level of education, age, height, weight, parity, as well as selected indicators of access to healthcare, food and nutrition, and mental health.

In addition, index variables will be included in the analysis to independently evaluate the impact of SARS-CoV-2 infection and lockdown during pregnancy on outcomes in childhood. The index variables can take four possible values depending on whether the pregnant woman was COVID-19 positive (c+) or negative (c−) and whether she was exposed (m+) or not (m−) to lockdown during pregnancy: c+m+, c+m−, c−m+, c−m−. Comparisons (ie, contrasts) of the posterior distributions for each of the four cases will enable us to assess the effect associated with each combination of exposures.

Furthermore, generalised mixed models will be used to evaluate the impact of COVID-19 infection on the women's biochemical measurements. These models will be adjusted by gestational age, maternal age, BMI, etc. COVID-19 infection status during pregnancy will be modelled as an index variable. The impact of COVID-19 on birth/pregnancy outcomes will be assessed by means of logistic regression.

## Patient and public partnership strategy

The study proposal was devised by stakeholders in India including regional government, hospital and public health officials. A range of community engagement meetings will take place before the recruitment of study participants to inform the communities about the study and its potential impact. The meetings will also provide an avenue to address any concerns or questions regarding the study protocol. Study participants and the public will be involved in the dissemination of the study's findings through community discussion or engagement events.

**Author affiliations**
[1] Department of Epidemiology and Population Health, London School of Hygiene and Tropical Medicine, London, UK
[2] Department of Epidemiology, Indian Institute of Public Health, Gandhinagar, India
[3] Clinical Division, ICMR-National Institute of Nutrition, Hyderabad, India
[4] Clinical Division, National Institute of Nutrition, Hyderabad, India
[5] Maternal and Child Health and Nutrition, National Institute of Nutrition, Hyderabad, India
[6] Department of Pathobiology and Population Sciences, University of London, London, UK
[7] Faculty of Children and Learning, University of London Institute of Education, London, UK
[8] Faculty of Children and Health, University of London Institute of Education, London, UK
[9] Department of Population Health, London School of Hygiene and Tropical Medicine, London, UK
[10] Department of Pathobiology and Population Sciences, London International Development Centre, London, UK

**Contributors** The author's contributions to writing the manuscript were as follows: MLJ wrote the first draft and was responsible for compiling the content of the manuscript for submission. CH, DBS and BK developed the concept of the study. MLJ, CH, DBS, BK, KS, PAS, JD, CA and SKB contributed to writing the original study protocol which forms the basis of this manuscript. MC, KS, TD, DPP, FM and RM contributed to development of the protocol and editing the manuscript. HD-K and EF developed the methods for anthropometry and dietary intake assessment and contributed to writing the manuscript. BC-U and MLJ developed and wrote the analysis plan. All authors contributed to the review of the manuscript for publication.

**Funding** The work was supported by UK Research and Innovation (UKRI) under its Global Challenge Research Fund (GCRF) (funding reference: MR/S01313X/1).

**Competing interests** None declared.

**Patient and public involvement** Patients and/or the public were not involved in the design, or conduct, or reporting, or dissemination plans of this research.

**Patient consent for publication** Parental/guardian consent obtained.

**Ethics approval** Approval for the study was obtained from the ethics committees of the State Health System Resource Centre, Gandhinagar, Gujarat, India (reference no: SHSRC/2021/2185), Indian Council of Medical Research (ICMR), National Institution of Nutrition, Hyderabad, India (reference no: EC/NEW/INST/2021/1206) and the ethics committee of London School of Hygiene and Tropical Medicine (reference no: 72848). The results of the study will be disseminated through public engagement meetings, presentation at conferences, seminars, meetings and publication in peer-reviewed, open-access journal. The results will also be deposited on an open-access data repository.

**Provenance and peer review** Not commissioned; externally peer reviewed.

**Data availability statement** No data are available.

**ORCID iDs**
Modou Lamin Jobarteh http://orcid.org/0000-0002-7350-6980
Santosh Kumar Banjara http://orcid.org/0000-0002-0893-9552
Priyanka Akshay Shah http://orcid.org/0000-0001-9970-6086
Farjana Memon http://orcid.org/0000-0003-3960-7929
Dharani Pratyusha Palepu http://orcid.org/0000-0003-1692-020X
Hilary Davies-Kershaw http://orcid.org/0000-0002-2044-2469

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
