## [Reviewer comments · BMJ Paediatrics Open]

ARTICLE DETAILS

TITLE (PROVISIONAL)	Impact of SARS-CoV-2 infection and mitigation strategy during pregnancy on prenatal outcome, growth and development in early childhood in India: a UKRI GCRF Action Against Stunting Hub protocol paper
AUTHORS	Jobarteh, Modou Saxena, Deepak B. Kulkarni, Bharati Shah, Komal Banjara, Santosh Kumar Shah, Priyanka Memon, Farjana Chilumula, Monica Palepu, Dharani Selvaraj, Kiruthika Dasi, Teena Madhari, Radhika Calvo-Urbano, Beatriz Dockrell, Julie Antalek, Catherine Davies-Kershaw, Hilary Ferguson, Elaine Heffernan, Claire

VERSION 1 - REVIEW

REVIEWER	Dr. Peter Flom Peter Flom Consulting
REVIEW RETURNED	07-Mar-2023

GENERAL COMMENTS	I confine my remarks to statistical and methodological aspects of this article. The general approach is fine, but I have rather a lot of suggestions and comments. But, this being a protocol, all these suggestions only amount to "minor revision". (This is one big reason why it is good to read protocol papers. I commend BMJ for publishing them.) And I thank the authors for adding their own, continuous line numbers. That makes writing the review easier. *General comments*: It would be good to be more specific about who you are recruiting. The pandemic has already lasted 3 years and isn't over. If you recruit women who got pregnant in Jan 2020 and also those who got pregnant in Jan 2023, then many of the measures will have to be different for the different kids, and that will make things very complex to analyze. On lines 111, this is further detailed. It is good to do
--

	matching, but that won't account for the different tests and measures. I suggest a narrower age range of the babies. Following the children for 18 months is good. Also, this is a very ambitious study! That's not bad or good, in itself, but it will take a lot of resources. *Specific comments* Lines 111 - 113 More details on the matching would be good. There are many methods of doing this. I don't have a strong opinion on which is best. Matching for the sex of the baby is pretty straightforward, but how will age matching be done? Also, why choose these three variables? They seem reasonable, but there are surely more variables that would be good. (E.g. economic status, age of parent, etc.) One method is to use a lot of variables and create a propensity score. You don't *have* to do this. But the methods should be detailed and (if possible) justified. Lines 114 -117 Good! Multiple timepoints allow better analysis. Line 127 and/or Table 1 More detail of some measures would be good; particularly the measures of cognition and other latent variables. It is quite tricky to measure the neurodevelopment of infants. This will affect power. Reading on, I see a lot of this on lines 235 ff. So, maybe this is OK. BMJ editors may have suggestions for where the details should be put. Line 165 HOW will this be done? There are lots of methods of tracking food intake. None are perfect. Please say what method(s) you will use. Line 280 I don't think it's necessary to give the STATA command, but, if you choose to keep it, it would be nice to put it in a different font, or maybe on a separate line, for ease of reading. Line 282 Why choose -1.5 for LAZ in the unaffected group? Is this based on data? It seems low to me, but if that's what the data show, then fine. Line 287 ff It seems like you are switching to R here. That's not wrong, but I'm just curious as to why you are using STATA for some analysis and R for others. I don't know STATA well, but R can do everything in this paper. Line 293 What does "relatively flat" mean here? Even though you haven't got any data yet, you can decide your priors now. So, if you don't want to use flat (uninformed) priors, then that's OK, but it has to be justified.
--	--

REVIEWER	Dr. Susy Joseph Government Medical College Thiruvananthapuram, Pediatrics
REVIEW RETURNED	12-Mar-2023

GENERAL COMMENTS

First of all, let me congratulate the authors for taking up this very relevant topic of factors affecting the growth and development in children of mothers infected with SARS-CoV-2 infection during pregnancy. However, please give clarification to the following queries:

1. In the title, 'pregnancy outcome' would be a better word instead of 'prenatal outcome'.

The title can be modified as 'Impact of SARS- CoV- 2 infection and mitigation strategy on pregnancy outcome, growth and development in early childhood in India'.

2. I understand from your protocol that criteria for diagnosing SARS-CoV -2 infection in pregnancy was RT PCR positivity. But there is no mention in the study regarding the time of positivity ie, whether in 1st, 2nd or 3rd trimester which could have an impact on pregnancy outcome and growth & development. Also, whether the pregnant women were infected with SARS – CoV -2 infection more than once is also not mentioned.

3. In the study design section, the sentence – 'Mother-baby pairs in the two groups will be recruited postnatally, between 0-24 months after birth and prospectively followed for 18 months period' – Does this mean that the children were followed up till 18 months after birth? Please clarify.

4. Matching was done with age, sex and location but not with family income, occupation, socioeconomic status and cultural habits which could also affect the outcome.

5. How was maternal mental health assessed? Please explain.

6. Regarding the questionnaire given to women – Was it a one-to-one session or group session? Whether the literacy of women could have affected their answers? How much time was given for the questionnaire?

7. Since there is a qualitative component to the study protocol, the study becomes a mixed study. So, the authenticity of results will be based on a properly addressed qualitative study which needs elaborate mention in the methodology session.

8. Whether ACE 2 & TMPRESS2 remain in blood for long time? Also, is there a definite relation between vitamin D, zinc and iron with these? Please cite literature.

9. Age at which children in the mother baby dyad are recruited into the study? Is it uniform in the two groups?

10. 'Measurement will be conducted in triplicate'. Please explain.

11. What is the validity and accuracy of CREDI & OX – NDA in detecting childhood neurocognitive development. Whether these criteria meet the standards of Indian children? Please cite relevant literature.

12. Please give the sequence and periodicity of assessments clearly in the methodology section.

13. This being a case control study, outcome is dependent on recall from mother. Can it bring bias in the study which can affect the final

	results? 14. Will the sample size calculation given in the protocol vary since it is a mixed study (both quantitative & qualitative). Why is Z score for height taken in calculating the sample size? Please explain. 15. Since it is an ambivalent study, the main study limitation would be the retrospective nature. Could any improvisation be done for this?
--	--

VERSION 1 – AUTHOR RESPONSE

Reviewer 1

I confine my remarks to statistical and methodological aspects of this article. The general approach is fine, but I have rather a lot of suggestions and comments. But, this being a protocol, all these suggestions only amount to "minor revision". (This is one big reason why it is good to read protocol papers. I commend BMJ for publishing them.)

And I thank the authors for adding their own, continuous line numbers. That makes writing the review easier.

***General comments*:**

It would be good to be more specific about who you are recruiting. The pandemic has already lasted 3 years and isn't over. If you recruit women who got pregnant in Jan 2020 and also those who got pregnant in Jan 2023, then many of the measures will have to be different for the different kids, and that will make things very complex to analyze. On lines 111, this is further detailed. It is good to do matching, but that won't account for the different tests and measures. I suggest a narrower age range of the babies.

Following the children for 18 months is good.

Also, this is a very ambitious study! That's not bad or good, in itself, but it will take a lot of resources.

Specific comments

Lines 111 - 113 More details on the matching would be good. There are many methods of doing this. I don't have a strong opinion on which is best.

Matching for the sex of the baby is pretty straightforward, but how will age matching be done? Also, why choose these three variables? They seem reasonable, but there are surely more variables that would be good. (E.g. economic status, age of parent, etc.) One method is to use a lot of variables and create a propensity score.

You don't *have* to do this. But the methods should be detailed and (if possible) justified.

Response: This study relies on hospital, community and birth records to recruit mother-baby pairs. Hospitals in parts of India have recorded the births of women who delivered during the pandemic, and also tested pregnant women for SARS-CoV-2 virus. These records will be used to recruit both the case and control groups. Where a case is identified, a control (a child with similar date of birth, sex and living within the same community) will also be identified and families approached through informed consent for recruitment. Whilst we agree that we cannot control for other equally important factors, we believe date of birth, sex and location are the critical factors to control without affecting the outcomes of the study. Other factors such as economic status, age of parent etc., will most definitely drive pregnancy and birth outcomes and even COVID-19 infectivity, thus controlling for those factors

in this study will be detrimental to the outcomes/findings. We have now provided some clarity around this. See lines 112-113 in the revised manuscript.

Lines 114 -117 Good! Multiple timepoints allow better analysis.

Response: We agree. Thanks for mentioning it.

Line 127 and/or Table 1 More detail of some measures would be good; particularly the measures of cognition and other latent variables. It is quite tricky to measure the neurodevelopment of infants. This will affect power. Reading on, I see a lot of this on lines 235 ff. So, maybe this is OK. BMJ editors may have suggestions for where the details should be put.

Response: We agree that estimating neurocognition in children is very difficult. We are using methods developed by the Intergrowth consortium led by the University of Oxford. The methods have been validated across 14 countries including India. The study staff to administer the assessments went through rigorous training by the developers of the methods. We are using objects which are familiar in India, and the assessments are conducted by local staff, who could relate with the participants to improve uptake and reliability of the assessment. Details of the methods are provided in the neurocognition sub-section, lines 238 – 261 in the revised manuscript. No further changes made to the manuscript.

Line 165 HOW will this be done? There are lots of methods of tracking food intake. None are perfect. Please say what method(s) you will use.

Response: We agree with the reviewer. We have now mentioned that a non-quantitative FFQ (food frequency questionnaire) will be used to track food intake during the pandemic. See lines 166-167 in the revised manuscript.

Line 280 I don't think it's necessary to give the STATA command, but, if you choose to keep it, it would be nice to put it in a different font, or maybe on a separate line, for ease of reading.

Response: We agree. We have changed the font to italics to facilitate comprehension. See line 284 of the revised manuscript.

Line 282 Why choose -1.5 for LAZ in the unaffected group? Is this based on data? It seems low to me, but if that's what the data show, then fine.

Response: This estimation is based on the mean (average) LAZ of non-stunted children under 5 years of age in the study community. No further changes made to the manuscript

Line 287 ff It seems like you are switching to R here. That's not wrong, but I'm just curious as to why you are using STATA for some analysis and R for others. I don't know STATA well, but R can do everything in this paper.

Response: STATA was used to estimate sample size. This is down to individual preference. R could possibly do a similar function. However, we are using the WHO R package to estimate LAZ and WLZ. This is a standard practice across similar studies and provide comparability of outcome. Additionally, the Bayesian networks analysis will be conducted on R. No further changes made to the manuscript

Line 293 What does "relatively flat" mean here? Even though you haven't got any data yet, you can decide your priors now. So, if you don't want to use flat (uninformed) priors, then that's OK, but it has to be justified.

Response: A flat distribution assigns the same probability to each value that a variable can take. This approach can be useful when we don't have a priori knowledge of the most likely values of the variable. However, on a revisit, we have opted to delete the statement, as it will give us the flexibility

to explore options for priors which are not flat. The statement has now been deleted from the revised manuscript.

Reviewer 2

First of all, let me congratulate the authors for taking up this very relevant topic of factors affecting the growth and development in children of mothers infected with SARS-CoV-2 infection during pregnancy. However, please give clarification to the following queries:

1. In the title, 'pregnancy outcome' would be a better word instead of 'prenatal outcome'.

The title can be modified as 'Impact of SARS- CoV- 2 infection and mitigation strategy on pregnancy outcome, growth and development in early childhood in India'.

Response: We thank the reviewer for this comment. Whilst we agree that 'pregnancy outcome' instead of 'prenatal outcome' might be better, we have gone really 'deep' with this title. It is the title of the study appended on ethics documents and other relevant study materials. Thus, we will be deeply grateful if the reviewer could please allow us to keep the name of the study as it is.

2. I understand from your protocol that criteria for diagnosing SARS- CoV -2 infection in pregnancy was RT PCR positivity. But there is no mention in the study regarding the time of positivity ie, whether in 1st, 2nd or 3rd trimester which could have an impact on pregnancy outcome and growth & development. Also, whether the pregnant women were infected with SARS – CoV -2 infection more than once is also not mentioned.

Response: We agree that the stage of pregnancy at which the women got infected with SARS-CoV-2 and whether they were re-infected during pregnancy are critical. We opted to not limit the criteria for inclusion of women based on stage of gestation at which they got infected and number of infections. However, data on the stage of pregnancy at which the women got infected, the variant of the virus and number of infections will be collected, analysed, and will contribute to the findings of the study.

3. In the study design section, the sentence – 'Mother-baby pairs in the two groups will be recruited postnatally, between 0-24 months after birth and prospectively followed for 18 months period' – Does this mean that the children were followed up till 18 months after birth? Please clarify.

Response: We planned to recruit mother-baby pairs at birth or soon after and follow them longitudinally for 18 months period. However, this is not always possible in field studies. As a result, we opted to widen the age of recruitment of babies. This will help us meet our recruitment targets without having to go through ethics amendment when the ideal situation of recruiting mother-baby pairs at birth or soon after is not met. We envisaged our approach will help us meet our recruitment targets without any major delays. All children recruited, irrespective of age, will be followed for 18 months period.

4. Matching was done with age, sex and location but not with family income, occupation, socioeconomic status and cultural habits which could also affect the outcome.

Response: We agree there might be other factors which also require controlling to facilitate unbiased comparison between the groups. However, we think date of birth, sex of child and location are factors that can be reasonably controlled among the study groups without having a negative impact on the study outcomes. Factors such as income, occupation, socioeconomic status and cultural practices are important risk factors of pregnancy/birth outcomes and COVID-19 infection, thus controlling them will not serve the purpose of the study.

5. How was maternal mental health assessed? Please explain.

Response: A qualitative questionnaire developed and validated will be used to assess the impact of the pandemic on maternal mental health. We have now provided text around this in the revised manuscript. See lines 184-187.

6. Regarding the questionnaire given to women – Was it a one-to-one session or group session? Whether the literacy of women could have affected their answers? How much time was given for the questionnaire?

Response: The questionnaires will be delivered in one-to-one session by trained enumerators. The staff will speak the same language as participants. Questionnaire will also be translated into local languages to facilitate accurate transmission of the questions. Administering the questionnaires could range between 10-30 minutes.

7. Since there is a qualitative component to the study protocol, the study becomes a mixed study. So, the authenticity of results will be based on a properly addressed qualitative study which needs elaborate mention in the methodology session.

Response: Our research is indeed somewhat a mixed method study. We are aware of this and its limitation.

8. Whether ACE 2 & TMPRESS2 remain in blood for long time? Also, is there a definite relation between vitamin D, zinc and iron with these? Please cite literature.

Response: ACE2 and TMPRSS2 are ubiquitously produced in humans performing various physiological functions. These functions are not only limited to SARS-CoV-2 infection. The relationship between nutritional status (minerals and vitamins) and SARS-CoV-2 infection is evolving with some evidence of relationship. Here are some interesting literature around it: <https://doi.org/10.1016/j.cdnut.2023.100044>, <https://doi.org/10.1016/j.clnu.2021.08.015> and <https://doi.org/10.1515/hmbci-2020-0074>. We would like to further investigate these links in our current study.

9. Age at which children in the mother baby dyad are recruited into the study? Is it uniform in the two groups?

Response: Mother-baby pairs will be recruited at birth or soon after. The age at recruitment is uniform in the two groups.

10. 'Measurement will be conducted in triplicate'. Please explain.

Response: A measurement will be conducted by an enumerator and repeated twice to obtain 3 measurements (triplicate) of the same parameter. We will use the triple measures to estimate intra and inter-rater reliability.

11. What is the validity and accuracy of CREDI & OX – NDA in detecting childhood neurocognitive development. Whether these criteria meet the standards of Indian children? Please cite relevant literature.

Response: CREDI and OX-NDA are fairly established questionnaires used to assess cognitive development in children. Developed by the Intergrowth consortium led by University of Oxford, OX-NDA has been validated in 14 countries including India. The study staff to administer the assessments went through rigorous training by the developers of the methods. We are using objects which are familiar in India, and the assessments are conducted by local staff, who can interact and relate with the participants to improve uptake and reliability of the assessment.

12. Please give the sequence and periodicity of assessments clearly in the methodology section.

Response: The frequency of the assessment is detailed in the method section. Table 1 also provides details. Assessments will be conducted at enrolment, 6, 12 and 18 months later.

13. This being a case control study, outcome is dependent on recall from mother. Can it bring bias in the study which can affect the final results?

Response: We agree that there are aspects of recall bias associated with some of the questionnaires such as dietary intake. We will clearly state this as a limitation of the study during publication of the study findings.

14. Will the sample size calculation given in the protocol vary since it is a mixed study (both quantitative & qualitative). Why is Z score for height taken in calculating the sample size? Please explain.

Response: Our primary outcome is child growth. Thus, it only makes sense to base the sample size calculation on Z score.

15. Since it is an ambivalent study, the main study limitation would be the retrospective nature. Could any improvisation be done for this?

Response: The retrospective nature of the study and associated recall bias are limitations of the study. We will acknowledge this in all future reporting of the study findings. At the moment there is nothing much we can do in this regard.

VERSION 2 – REVIEW

REVIEWER	Dr. Peter Flom Peter Flom Consulting
REVIEW RETURNED	15-Apr-2023

GENERAL COMMENTS	The authors have addressed my concerns and I now recommend publication.
---

REVIEWER	Dr. Susy Joseph Government Medical College Thiruvananthapuram, Pediatrics
REVIEW RETURNED	07-May-2023

GENERAL COMMENTS	The authors have clarified all points with the needed literature references. But the major limitation is the retrospective nature of the study which has to be mentioned as the major study limitation. Also, the recruitment of babies are not done uniformly from birth for follow up up to 18 months.
--

VERSION 2 – AUTHOR RESPONSE

Reviewer 2

The authors have clarified all points with the needed literature references. But the major limitation is the retrospective nature of the study which has to be mentioned as the major study limitation. Also, the recruitment of babies are not done uniformly from birth for follow up up to 18 months.

Response: We agree with the reviewer's views on the limitation of the study. The study does not have the resources to set up a prospective pregnancy cohort, where women are recruited before they became infected with COVID-19 during pregnancy and longitudinally follow children of COVID-19 positive and negative women within the cohort. Instead, we recruited women who were pregnant during the pandemic had COVID-19 infection (and controls with no known COVID-19 infection) during pregnancy, and longitudinal follow their children. Although this was the best approach we could take, we will highlight the limitations of our approach in all future publications. In general, the mother-baby pairs were recruited at birth or soon after.